# When Albumin Meets Liposomes: A Feasible Drug Carrier for Biomedical Applications

**DOI:** 10.3390/ph14040296

**Published:** 2021-03-26

**Authors:** Kazuaki Taguchi, Yuko Okamoto, Kazuaki Matsumoto, Masaki Otagiri, Victor Tuan Giam Chuang

**Affiliations:** 1Faculty of Pharmacy, Keio University, 1-5-30 Shibakoen, Minato-ku, Tokyo 105-8512, Japan; taguchi-kz@pha.keio.ac.jp (K.T.); matsumoto-kz@pha.keio.ac.jp (K.M.); 2Faculty of Pharmaceutical Sciences, Sojo University, 4-22-1 Ikeda, Nishi-ku, Kumamoto 862-0082, Japan; nkhymru7@gmail.com (Y.O.); otagirim@ph.sojo-u.ac.jp (M.O.); 3DDS Research Institute, Sojo University, 4-22-1 Ikeda, Nishi-ku, Kumamoto 862-0082, Japan; 4School of Pharmacy and Biomedical Sciences, Curtin University, Perth, WA 6102, Australia

**Keywords:** albumin, liposome, drug delivery, enhanced permeability retention effect, protein

## Abstract

Albumin, the most abundant protein in plasma, possesses some inherent beneficial structural and physiological characteristics that make it suitable for use as a drug delivery agent, such as an extraordinary drug-binding capacity and long blood retention, with a high biocompatibility. The use of these characteristics as a nanoparticle drug delivery system (DDS) offers several advantages, including a longer circulation time, lower toxicity, and more significant drug loading. To date, many innovative liposome preparations have been developed in which albumin is involved as a DDS. These novel albumin-containing liposome preparations show superior deliverability for genes, hydrophilic/hydrophobic substances and proteins/peptides to the targeting area compared to original liposomes by virtue of their high biocompatibility, stability, effective loading content, and the capacity for targeting. This review summarizes the current status of albumin applications in liposome-based DDS, focusing on albumin-coated liposomes and albumin-encapsulated liposomes as a DDS carrier for potential medical applications.

## 1. Introduction

Albumin is a multifunctional plasma protein that constitutes about 60% of the total plasma proteins [1]. Albumin purified from the human plasma fraction was used for the first time as a plasma expander to treat hemorrhaging patients during World War II. Albumin infusions are now essential products for treating hemorrhages and refractory ascites associated with chronic liver cirrhosis, hepatorenal syndrome, and spontaneous bacterial peritonitis [2,3,4]. In addition to its use for therapeutic purposes, quantitative and qualitative endogenous changes in albumin are important laboratory data that contribute to modern medical care. For example, a reduction in plasma albumin levels can be an indicator of several possible diseases, such as liver cirrhosis and malnutrition. The posttranslational modification of the cysteine residue at position 34 (^34^Cys), the only unpaired cysteine residue in the albumin molecule, can be used to monitor the severity of renal and liver disorders [5]. For highly protein-bound drugs, such as teicoplanin and phenytoin, dosage adjustments are recommended based on plasma albumin levels [6,7].

With the advancements in biotechnology and protein science, many albumin-associated pharmaceutical preparations are currently on the market, such as nab-paclitaxel (Abraxane^®^), albumin nanoparticles prepared by nab technology [8]. Nab-paclitaxel (Abraxane^®^) makes it possible for paclitaxel to be delivered in a soluble form without the need for a solubilizing agent (polyoxyethylated castor oil; Cremophor^®^), which is used in a conventional paclitaxel preparation (Taxol^®^). A clinical benefit of using albumin as an alternative to Cremophor^®^ is the decreased severe side effects, such as hypersensitive reactions. Albumin also facilitates transcytosis across the endothelial barrier, thus leading to the high distribution of paclitaxel in the region of the cancer via the albumin receptor (gp60) [8]. Another example of an albumin product is etelcalcetide (Parsabiv^®^), a linear heptapeptide preparation that is used for treating secondary hyperparathyroidism in chronic kidney disease patients who are on hemodialysis. This product has a long duration of action due to the presence of a disulfide bond between albumin and etelcalcetide [9]. In addition to nab-paclitaxel (Abraxane^®^) and etelcalcetide (Parsabiv^®^), other pharmaceutical preparations, such as Levemir^®^ and Victoza^®^ etc., also involve the use of albumin-associated preparations [10]. These preparations take advantage of the following inherent structural and physiological characteristics of albumin—Albumin has a good biocompatibility with a long plasma half-life, the aqueous solubility of endogenous and exogenous hydrophobic substances can be improved by their binding to albumin, and the cysteine residue at position 34 of albumin readily reacts with oxidative products and other thiol groups (Figure 1).

Both academia and industry have used various approaches to develop novel albumin-associated pharmaceutical preparations: (i) albumin-drug conjugation to prolong blood retention [11,12,13,14,15], (ii) nanoparticulation by albumin aggregates to improve stability and the loading efficiency of hydrophobic substances [16,17,18], (iii) liposome, micelles, niosomes, and emulsions fused with albumin to compensate for their drawbacks [19,20,21,22,23]. Among these approaches, adapting albumin for use in a liposome-based drug delivery system (DDS) has shown promising results. The efforts in this area include modifications of the liposome surface with albumin replacing polyethylene glycol (PEG) or a ligand for organ targeting [24,25], and the encapsulation of albumin together with bound hydrophobic substances into the aqueous liposome core to improve the encapsulation efficiency of hydrophobic substances [26]. This review summarizes the current status of using albumin in liposome-based DDS.

## 2. Liposome Coated with Albumin

Albumin is a dysopsonized protein. Coating liposomes with albumin molecules is an approach for delivering a drug by pharmacokinetic modification, mainly to extend the plasma half-life of the drug, via modification of opsonization and recognition by macrophages [27,28,29]. Depending on the desired outcome of drug delivery, the modification of any one or both of the two main components, i.e., albumin and liposomes, can be employed to adjust the structural and physicochemical characteristics of a system and therefore the corresponding pharmacokinetic properties of the system.

Two types of albumin-coated liposomes have been reported—A liposome that contains albumin that is linked to a polymer, or albumin adsorbed on the surface of a cationic liposome via electrostatic interactions (Figure 2). This section reviews our current state of knowledge concerning albumin-coated liposomes as a DDS carrier, focusing on the benefits of using albumin-coated liposomes in biomedical applications.

### 2.1. Methods for Preparing Albumin-Coated Liposomes

The most commonly used method for accomplishing this is to modify the surface of cationically charged liposomes with anionically charged albumin, resulting in electrostatic interactions at pH values higher than the isoelectric point (pI) of albumin. To prepare albumin-coated cationic liposomes, albumin in a solution of phosphate-buffered saline (pH 7.4, 1:1, *v*/*v*) was drop-wise added to the prepared positively charged liposomes at 37 °C over a 1 h period, and the albumin-coated cationic liposomes were isolated by centrifugation [30]. Encapsulating unmodified albumin by inducing flocculation provides a drug delivery vehicle with a combination of the versatility of a liposome and the drug-binding ability of albumin. An alternative approach is to incubate the mixture at 4 °C to induce flocculation with the pH of the mixture adjusted to 6.5, and then place it in a slow-shaking bath for 20 h before centrifugation at 15,000 rpm for 20 min [31]. The entrapment efficiency of vancomycin, a model water-soluble drug with moderate albumin-bound properties, was influenced by the protein concentration and the initial charge on the liposome. 

Another method for preparing albumin-coated liposome is to modify the surface of the liposome by an albumin–polymer conjugation. In most cases, albumin was covalently conjugated with a lipid-polymer, such as 1,2-distearoyl-*sn*-glycero-3-phosphoethanolamine-poly(ethylene glycol) (DSPE-PEG), via a coupling agent. For example, Award et al. succeeded in attaching albumin to DSPE-PEG via 2,4,6 trichloro-1,3,5 triazine, in which the lysine residues of albumin were covalently conjugated with the amino group of DSPE-PEG [32]. In addition to using 2,4,6 trichloro-1,3,5 triazine, it is also possible to conjugate DSPE-PEG-maleimide to the cysteine residue of albumin, thus modifying the surface of the liposome [33]. A unique approach that does not use a coupling agent was recently reported [25]. The liposome contained DSPE-PEG with an alkyl terminal as a component of the lipid membrane that was prepared and subsequently mixed with albumin. The as-prepared liposome was coated with albumin on a PEGylated liposome surface because albumin specifically binds the alkyl ligand via the hydrophobic pockets on the albumin molecule. 

### 2.2. Extending Plasma Half-Life and Decreasing Toxicity

Non-specific interactions between nanoparticles and plasma proteins occur when the nanoparticles enter the blood. As a result, coating the original nanoparticles with a protein corona alters the in vivo fate of nanoparticles as well as the biological responses to the nanoparticles. This is one of the substantial reasons for the two main problems associated with nanoparticle-based drug delivery systems, i.e., nanotoxicity and the rapid clearance of nanoparticles from the blood after intravenous injection. A preformed albumin corona has been shown to inhibit plasma proteins such as IgG adsorption, decrease complement activation, reduce the macrophage-related phagocytosis, and ultimately prolong the blood circulation time and reduce the toxicity of the nanoparticles. Furthermore, nanoparticle-albumin complexes show an enhanced bioavailability compared to naked nanoparticles [34].

Liposomes are plagued by rapid opsonization, leading to the shortening of their circulation time in the bloodstream. Interactions between liposomes and plasma proteins have been investigated in order to evaluate the role of liposomal lipid composition and concentration in forming the protein corona. Protein corona formation depends on the presence of PEG, cholesterol, the concentration of the components, and surface charge. Liposomes incorporating 1,2-dipalmitoyl-*sn*-glycero-3-phosphocholine lipids and cholesterol both in a molar ratio of 11% were reported to be stable over time, while their size was not altered dramatically in a biological medium. Liposomes containing cholesterol and PEGylated lipids retain their size in the presence of serum as well as their physical stability. Lipid composition and concentration have an effect on the adsorption of proteins and on liposomal stabilization [35].

The encapsulation of curcumin in a liposome improves its bioavailability and stability, but the short half-life remains an issue. Albumin was used to coat the liposomal curcumin to produce a long-circulating delivery system. Albumin-coated liposomal curcumin particles were more spherical, more homogeneous in size, and significantly larger than liposomal curcumin. The presence of albumin can enhance the stability of the liposome structure and slow down the release of curcumin. An albumin coating resulted in a considerable reduction in the macrophage phagocytosis of liposomal curcumin [22].

### 2.3. Biomedical Applications of Albumin-Coated Liposome

#### 2.3.1. Oral and Vaginal Medications

The oral delivery of peptide/protein drugs has always encountered the gastrointestinal tract’s rigorous defenses, mainly mucus and epithelium barriers. Liposomes coated with albumin are a promising approach to increasing the oral delivery of peptides/proteins. Wang et al. reported that albumin adsorbed to cationic liposomes to form albumin-coated liposomes showed 3.24-fold higher uptake and a 7.91-fold higher transepithelial permeability than free insulin. An albumin corona can be shed from albumin-coated liposomes as they cross the mucus layer, resulting in the exposure of the cationic liposomes, thus improving transepithelial transport. Furthermore, albumin-coated liposomes showed up to an 11.9% oral bioavailability in type I diabetic rats with remarkable hypoglycemic effects [36].

Itraconazole-loaded nanoparticles containing albumin and liposomes can be prepared by a technological process that avoids the use of organic solvents and crosslinking agents. The opposite charges of albumin and liposomes combined with a low temperature enable the protein and vesicles to flocculate with the formation of particles entrapping the liposomes and itraconazole-bound (drug entrapment efficiency: 51–68%) and unbound albumin. Mannitol was used as a lyoprotectant, and the freeze-dried cake was directly compressed into tablets for vaginal administration. In vitro drug delivery assays showed that the particles with higher albumin amounts were released more rapidly than those with lower albumin contents. Furthermore, the results of this study demonstrated that albumin particles entrapping liposomes are a green pharmaceutical vehicle with a high potential for delivering hydrophobic and highly albumin-bound drugs [37].

#### 2.3.2. Gene Delivery

Interactions with biological macromolecules in the blood and tissues is a significant drawback to the use of highly positively charged cationic liposome-DNA complexes (“lipoplexes”) that have been developed as gene delivery vehicle systems. Covering the lipoplex surface with negatively charged albumin would be expected to mask the high zeta potential of the lipoplex, resulting in a decreased toxicity. Furthermore, coating a lipoplex with albumin resulted in a significantly higher gene expression in the lungs and spleen of mice and conferred resistance to inhibition by serum components [38]. In addition, the hepatic and pulmonary gene expression levels were increased in the albumin-coated lipoplex compared to a naked lipoplex in hepatitis mice induced by carbon tetrachloride [39]. These facts indicate that coating a lipoplex with albumin is a feasible approach for ameliorating lipoplex toxicity without inhibiting transfection activity.

The prognosis of acute myeloid leukemia remains poor despite the advances made in chemotherapy and bone marrow transplantations. The narrow tissue specificity of folate receptor β (FRβ) that is aberrantly expressed in hematological linage cell lines has been targeted for drug delivery. To utilize the characteristics of albumin-coated liposomes, such as high safety and transfection activity, an albumin-coated lipoplex modified with folic acid was applied to the delivery of siRNA to myeloid cells. siRNA-loaded cationic liposomes coated with folic acid-modified albumins could help stabilize the delivery system in the circulation and achieve an FRβ-mediated active targeting effect to acute myeloid leukemia. Since all-trans retinoic acid (ATRA) has been reported to selectively upregulate the expression of FRβ, the systematic administration of ATRA was found to significantly promote endocytosis. It increases the intracellular concentration of the albumin-coated lipoplex modified with folic acid. This strategy combines the FRβ amplification effect with siRNA’s effective delivery, a most desirable action for acute myeloid leukemia-targeting therapy [40].

#### 2.3.3. Liver Targeting

Liver fibrosis progression involves hepatic stellate cells (HSCs) that produce a secreted protein that is acidic and rich in cysteine (SPARC), a specific protein-binding protein that binds strongly to albumin. Wang et al. prepared naringenin-loaded albumin-coated liposomes for use in the delivery of naringenin to the liver. Naringenin is a specific Smad3 inhibitor that blocks the TGF-β/Smad3 signaling pathway to activated HSCs, thus having an anti-fibrosis effect. The albumin-coated surface of the liposomes significantly reduced their aggregation and naringenin leakage and increased their stability. Furthermore, the uptake of naringenin-loaded albumin-coated liposomes by activated HSCs was 1.5 times higher than that of naringenin-loaded liposomes with no albumin coating, suggesting that naringenin-loaded albumin-coated liposomes specifically increased the targeting of activated HSCs via albumin and SPARC-dependent pathways in treating liver fibrosis. These results indicate that albumin-coated liposomes would be a feasible DDS carrier to SPARC-expressing cells such as HSCs and pancreatic cancer [41].

## 3. Albumin-Encapsulated Liposome

Numerous attempts have been made to develop liposomes that encapsulate proteins, such as hemoglobin, erythropoietin, and lactoferrin, into the core of the liposome [42,43,44,45]. The primary purposes of encapsulating these proteins into liposomes are to improve the delivery of proteins to the targeting area and prolong the blood retention of proteins and low molecular weight compounds by loading them into a liposome. On the other hand, the principal purpose for encapsulating albumin into a liposome is to permit hydrophobic substances to be encapsulated into the aqueous core of liposomes in the form of an albumin-drug complex [46,47]. This approach is limited to hydrophobic substances that have a high binding affinity for albumin. However, this strategy has an advantage over the standard approach for incorporating hydrophobic substances into the lipid bilayer of a liposome. Incorporating a hydrophobic drug into a liposome’s lipid membrane leads to significantly decreased long-term stability, and only limited amounts can be loaded [48].

As of this writing, two types of albumin-encapsulated liposomes have been reported—A liposome that encapsulates an albumin-drug complex or a drug-loaded albumin nanoparticle (Figure 3). In addition to the stable encapsulation of significant amounts of hydrophobic substances, albumin-encapsulated liposomes also permit the co-loading of another hydrophobic substance with a low albumin binding into the liposomal membrane. For example, Ruttala and Ko prepared albumin-encapsulated liposomes that contain embedded curcumin, a hydrophobic drug with a low albumin binding affinity (curcumin: K = 2.52 × 10^5^ M^−1^ [49]) into the liposomal membrane, and encapsulated albumin complexed with paclitaxel, a hydrophobic drug with a high albumin binding affinity (paclitaxel: K = 1.7 × 10^6^ M^−1^ [50]) into the aqueous core of a liposome [51]. In addition, Yu et al. modified the surface of liposome-encapsulated albumin nanoparticles containing paclitaxel with both an enzyme-responsive peptide (fibroblast-activated protein-a responsive cleavable amphiphilic peptide) and a photothermal agent (IR-780) to increase the therapeutic efficacy of the system against pancreatic tumors [52].

### 3.1. Methods for the Preparation of Albumin-Encapsulated Liposomes

Three reported methods for preparing albumin-encapsulated liposomes are (1) the thin-film hydration method, (2) the reverse-phase evaporation method, and (3) the ethanol injection method (Figure 4). This section provides a summary of these methods.

#### 3.1.1. Thin-Film Hydration Method

The most commonly used method for preparing albumin-encapsulated liposomes is the thin-film hydration method. In this method, a lipid film with optimal compositions is prepared in a round-bottom flask under aspirator vacuum conditions. The preparation is subsequently hydrated by adding an aqueous solution containing an albumin-drug complex or a drug-loaded albumin nanoparticle. However, the liposomes prepared by this method are multilamellar vesicles, resulting in a low albumin encapsulation efficiency due to the small volume of the aqueous core of the liposome.

The freeze–thaw technique can improve the encapsulation efficacy of an albumin-drug complex (or a drug-loaded albumin nanoparticle). Repetitive freeze–thaw cycles result in the membrane of the multilamellar vesicles being penetrated via ice crystal formation during the freezing process, thus disrupting the narrowly spaced lamellae of the liposomes. Consequently, the freeze–thaw process increases the ratio of the aqueous phase to the lipid phase, leading to a higher encapsulation efficiency [53]. Zhao et al. reported an increase of protein encapsulation efficiency (recombinant human growth hormone) based on the number of freeze–thaw cycles [54]. Lastly, the extrusion method is an optional tool that permits the preparation of liposomes with a targeted size.

#### 3.1.2. Reverse Phase Evaporation Method

The reverse-phase evaporation method [55] involves the use of a water-in-oil emulsion, which constitutes an organic phase (containing the lipids) and an aqueous phase (containing the albumin-drug complex) and is prepared by mixing and sonicating the two solutions. Albumin-encapsulated liposomes are then obtained by sonication, evaporation of the organic solvent, hydration, and centrifugation. This method produces large unilamellar vesicles. Thus, the extrusion method can be used to produce an optimal sized particle with a narrow size distribution. The reverse-phase evaporation method also results in a higher encapsulation efficiency of substances compared to the thin-film hydration method due to the high ratio of aqueous phase to lipid phase compared to multilamellar liposomes [56]. However, the encapsulation of the albumin-drug complex during the preparation process may cause albumin denaturation by sonication. The albumin-drug complex in the aqueous phase can contact an organic solvent, which could also lead to albumin denaturation.

#### 3.1.3. Ethanol Injection Method

The ethanol injection method is a straightforward approach for producing albumin-encapsulated liposomes [57]. An ethanol solution of a mixture of lipid and albumin is rapidly injected into an aqueous solution, which serves as a dispersant. This method is more straightforward, rapid, easier to scale-up, and does not require any special equipment compared to the other methods [58]. However, similar to the reverse-phase evaporation method, albumin denaturation and drug release are possible when the albumin-drug complex comes into contact with the ethanol. Furthermore, residual ethanol in the prepared liposome suspension forms an azeotrope with water [59], complicating in vivo applications from a safety perspective.

#### 3.1.4. Surface Modification for Active Targeting

Surface modification by ligands such as peptides and antibodies are commonly done for liposomes to enable active targeting [60]. As of this writing, there is no report of the preparation of surface-modified albumin-encapsulated liposomes for active targeting. Since various types of PEG can have various functionalized groups (–COOH, –SH, –NH_2_, –NHS, –CHO) [61,62], it would be the most useful method for covalently binding ligands to functionalized group-terminated PEG on albumin-encapsulated liposomes.

### 3.2. Factors Influencing Physicochemical Characteristics of Albumin-Encapsulated Liposomes

The physicochemical characteristics of a liposome, such as surface charge, diameter, and entrapment efficacy of drugs or albumin, are controlled by various liposome preparation regimens. The following factors appear to have an impact on their physicochemical properties.

#### 3.2.1. Preparation Method

The methods used to prepare liposomes can influence the physicochemical properties of albumin-encapsulated liposomes, primarily particle diameter, the structure of the encapsulated albumin, and the encapsulation efficiency for albumin. As described in Section 3.1, the thin-film hydration method has some drawbacks, in that the liposomes that are produced have a small aqueous core space compared to those produced by the reverse-phase evaporation method and the ethanol injection method. A higher encapsulation efficiency for albumin into liposomes was observed in the reverse-phase evaporation method [63] and the ethanol injection method [57] compared to the thin-film hydration method [64]. On the other hand, albumin denaturation is unavoidable when an organic solvent is used for reverse-phase evaporation and ethanol injection. In addition, liposomes formed by the above methods are generally heterogeneous in size. Thus, the extrusion method and sonication are useful for size control and size reduction, respectively.

#### 3.2.2. Lipid Membranes

The effects of lipid composition and the ratio of phospholipids and cholesterol have been reported to affect the physicochemical characteristics of liposomes that contain encapsulated albumin. Cholesterol is frequently used as a component of liposomal bilayers to regulate the permeability and rigidity of the liposomal membrane. Cholesterol in the liposome may affect the particle size and the encapsulation efficiency for albumin. Okamoto et al. reported that the diameter of albumin-encapsulated liposomes decreased with increasing molar ratio of cholesterol to phosphatidylcholine [64]. In addition, the efficiency of albumin encapsulation into liposomes decreases with increasing molar ratio of cholesterol to phosphatidylcholine [64,65]. Accordingly, the rigidity and fluidity of the lipid bilayer may be associated with the diameter and encapsulation efficiency of the albumin being used.

In addition to the surface charge of a liposome, charged lipids are also considered to be factors that affect the entrapment of albumin into a liposome. Based on a study reported by Liu et al., the encapsulation efficiency of lactoferrin (a positively charged protein) and albumin (a negatively charged protein) in anionic liposomes was approximately 46.7% and 33.6%, respectively [66,67]. Furthermore, Wang et al. reported that liposomes that are cationized by the presence of 1,2-dioleoyl-3-trimethylammonium-propane had a higher encapsulation efficiency for albumin compared to neutral liposomes [68]. These reports suggest that the encapsulation efficiency for a protein increases when it is encapsulated into a liposome with an oppositely charged lipid membrane due to electrostatic interactions between the protein and the liposomal membrane. However, there are exceptions to this rule. Liposomes cationized by the use of 1,2-dioleoyl-3-dimethylammonium-propane had a lower encapsulation efficiency for albumin compared to neutral liposomes [65]. Dai et al. reported that liposomes that are anionized by dioleoyl phosphatidyl glycerol had a higher encapsulation efficiency for albumin than neutral liposomes [63]. Therefore, the effects of charged lipids on the encapsulation efficiency of albumin into a liposome do not appear to follow a uniform pattern.

PEG-derivatized lipids are frequently used to modify the lipid surface of a liposomal membrane to increase the stability of the preparation in a vial and to provide a stealth effect in vivo [69,70]. Because PEG forms a layer on the outside, and inside of the lipid bilayer, the PEG chain may hinder the incorporation of albumin into the aqueous core. Reports have appeared in which the encapsulation efficacy of albumin between liposomes with and without PEG was compared [64,65]. Although the presence of PEG reduced the extent of encapsulation of albumin in the liposome, the extent of this reduction was within a narrow percentage, thus making it negligible.

#### 3.2.3. Buffer Solution

The ionic strength of a buffer is a crucial factor in preparing protein-encapsulated liposomes. For example, the entrapment of ovalbumin (a negatively charged protein) into cationic liposomes was higher in a buffer with a low ionic strength than one with a high ionic strength [71]. Wang et al. reported that the amount of albumin entrapped into neutral liposomes increased with increasing ionic strength. Interestingly, the amount of entrapped albumin was dramatically decreased when the ionic strength reached a concentration that was three times higher than the albumin concentration [65]. The exact mechanism responsible for this is unclear but may involve changes in electrostatic interactions between albumin and the lipid membrane.

### 3.3. Biological Properties of Albumin-Encapsulated Liposomes

#### 3.3.1. Biocompatibility

Although it is generally accepted that albumin and liposomes are ideal biomaterials with a high biocompatibility for drug carriers [72,73,74,75], the biocompatibility of albumin-encapsulated liposomes is not guaranteed due to the fact that contaminants, especially organic solvents used in the process of preparation can possibly be retained. The biocompatibility of albumin-encapsulated liposomes was investigated by Okamoto et al. [64]. In this in vitro study, whole blood with added albumin-encapsulated liposomes had no effect on blood cells (white blood cells, red blood cells, and platelets) as observed at 24 h. Furthermore, albumin-encapsulated liposomes did not affect the hepatic and renal functions of healthy mice. These results indicate that albumin-encapsulated liposomes have a favorable hemocompatibility and biocompatibility as a drug carrier. However, further safety data regarding albumin-encapsulated liposomes will be required.

#### 3.3.2. Pharmacokinetic Properties

Only a few studies have reported on the pharmacokinetics of albumin-encapsulated liposomes. Zhang et al. compared the plasma retention between chlorambucil, chlorambucil-loaded albumin nanoparticles, and chlorambucil-loaded albumin nanoparticle-encapsulated liposomes in rats [76]. The blood retention of the chlorambucil-loaded albumin nanoparticle-encapsulated liposomes was greater than that for chlorambucil only and the chlorambucil-loaded albumin nanoparticles, indicating that encapsulation into liposome provides superior blood retention effects. Similar results were reported by Wei et al. and Ruttala et al., who compared blood retention between paclitaxel, paclitaxel-loaded albumin nanoparticles, and paclitaxel-loaded albumin nanoparticle-encapsulated liposomes [77,78]. The longer blood retention effect of the albumin-encapsulated liposomes could be due to the stealth effect conferred by PEG modification that prevented accumulation in the liver [79]. Albumin-encapsulated liposomes were mainly distributed to the liver and spleen [64,78], suggesting that albumin-encapsulated liposomes were ultimately captured by mononuclear phagocyte systems, such as Kupffer cells and splenic macrophages, and are then metabolized and excreted similar to endogenous albumin and lipids. However, information on the pharmacokinetic properties regarding the potential for bioaccumulation, drug–drug interactions, and accelerated blood clearance, which are factors in liposome preparations, is currently not available [80,81,82].

Albumin-encapsulated liposomes were developed to deliver drugs to solid tumors, such as melanoma [76,78], breast cancer [83], pancreatic ductal adenocarcinoma [52,77], and pancreatic cancer [84]. The accumulation of albumin-encapsulated liposomes in an inflammatory site in the coon using a mice model of colitis has also been evaluated [85]. The primary strategy for the delivery of albumin-encapsulated liposomes is based on the concept that liposomes with a size of ~200 nm can passively accumulate into the tumor and the inflammatory site (Figure 5), a process that is referred to as the enhanced permeability retention (EPR) effect [86,87]. As expected, albumin-encapsulated liposomes successfully accumulated in solid tumors and inflammatory sites in all previously reported studies. Albumin-encapsulated liposomes accumulated at a higher level in melanoma [76,78] and pancreatic ductal adenocarcinoma [52,77] compared to albumin nanoparticles. This superior accumulation of albumin-encapsulated liposomes into the tumor is thought to contribute to the high antitumor and anti-inflammatory efficacies of these systems, as described below.

### 3.4. Biomedical Applications of Albumin-Encapsulated Liposomes

The application of albumin-encapsulated liposomes for use as anticancer drugs or as carriers of immunosuppressants is expected to be used to access solid tumors and inflammatory sites via the EPR effect. Several studies have reported on the therapeutic efficacy of drug-loaded albumin-encapsulated liposomes for the treatment of various types of solid cancers and colitis (Table 1).

#### 3.4.1. Pancreatic Cancer

There is no doubt that nab-paclitaxel (Abraxane^®^) is a key drug for treating pancreatic cancer [89,90]. Based on this fact, numerous attempts have been made to investigate the therapeutic potential of paclitaxel-loaded albumin-encapsulated liposomes against intractable pancreatic cancer in tumor-bearing mice. Okamoto et al. reported that a paclitaxel-albumin complex-encapsulated liposome greatly suppressed the growth of AsPC-1 human pancreatic cancer compared to nab-paclitaxel (Abraxane^®^) in xenograft tumor-bearing mice [84]. However, it is known that the pathophysiological characteristics of pancreatic cancers, which include fibrotic stroma and the production of pancreatic stellate cells, inhibit the penetration of antitumor agents, including nanoparticles, into the core of the cancer, resulting in reducing the response rate of chemotherapy [91,92]. Hence, it would be desirable to develop albumin-encapsulated liposomes that could facilitate the extent of penetration into this fibrotic extracellular matrix in pancreatic cancer. To achieve a high penetration of paclitaxel into a tumor core, Wei et al. co-encapsulated paclitaxel-loaded albumin nanoparticles and ellagic acid (an anti-pancreatic stellate cell agent)-loaded albumin nanoparticles into liposomes [77]. As a result, liposome co-encapsulated paclitaxel-loaded albumin nanoparticles and the ellagic acid-loaded albumin nanoparticles both penetrated deeply into the core of BxPC-3 (human pancreatic cells) and HPaSteC (human pancreatic stellate cells) tumors in xenograft tumor-bearing model mice, indicating that the co-encapsulation of ellagic acid represents a promising strategy for improving penetration into the matrix. Similar to the superior tumor permeability, tumor growth was dramatically decreased in xenograft BxPC-3 and HPaSteC tumor-bearing model mice that had been treated with liposome co-encapsulated paclitaxel-loaded albumin nanoparticle and ellagic acid-loaded albumin nanoparticles compared to all other reference groups.

Another group applied photothermal therapy using a near-infrared laser as a strategy to enhance the deep penetration of liposomes in pancreatic cancer [52]. This unique strategy focused on inducing pathohistological changes by photothermal therapy, which expands the interstitial tumor space by strong hyperthermia, leading to the deep penetration of macromolecules into the tumor. To equip this system with albumin-encapsulated liposomes, they embedded IR-780, a photothermal agent, into the lipid membrane of paclitaxel-loaded albumin nanoparticle-encapsulated liposomes. In vivo studies showed that near-infrared laser irradiation enhanced the penetration of the liposomes into the tumors of heterotopic Pan02 (mouse pancreatic cancer) and NIH3T3 (mouse embryonic fibroblast) bearing mice. In addition, the use of paclitaxel-albumin nanoparticle-encapsulated liposomes in photothermal therapy dramatically decreased the tumor volume and prolonged the duration of survival in orthotopic Pan02 pancreatic cancer-bearing mice compared to all other reference treatments, including paclitaxel-loaded albumin nanoparticles. These results indicate that albumin-encapsulated liposomes may be a potent and superior paclitaxel carrier compared to nab-paclitaxel (Abraxane^®^) for the treatment of pancreatic cancer.

#### 3.4.2. Breast Cancer

Since nab-paclitaxel (Abraxane^®^) has also been effectively used in the chemotherapy of breast cancer in clinical situations [93,94], the therapeutic potential of paclitaxel-loaded albumin-encapsulated liposomes has been examined. Ruttala and Ko reported that paclitaxel-loaded albumin nanoparticle-encapsulated liposomes exerted an increased cytotoxicity against 2D-cultured MCF-7 breast cancer cells compared to paclitaxel-loaded albumin nanoparticles [51,78]. Okamoto et al. also reported that paclitaxel-albumin complex-encapsulated liposomes had a potent antitumor effect on 2D-cultured and 3D-cultured triple-negative breast cancer cells (MDA-MB-231 cell), an intractable form of breast cancer that is resistant to conventional hormone and molecular target agent therapy due to the absence of progesterone receptors, estrogen receptors, and the human epidermal growth factor receptor-2 [88]. These results indicate that paclitaxel-loaded albumin-encapsulated liposomes have a potent anti-proliferative effect against breast cancer, including triple-negative breast cancer.

Liang et al. applied albumin-encapsulated liposomes to the internal radioisotope therapy of breast cancer [83]. In this strategy, ^131^I-labeled albumin-encapsulated liposomes, a β-ray emitting agent, facilitated tumor blood vascular permeability, resulting in an enhanced passive tumor accumulation of subsequently administered macromolecules. A ^131^I-labeled albumin-encapsulated liposome pretreatment potentiated the antitumor effect of second-wave immune checkpoint blockade therapy based on the anti-programmed death-ligand 1 antibody, hypoxia-activated chemotherapy based on AQ4N (hypoxia-activatable drug) loaded liposomes, and photothermal therapy based on near-infrared dye loaded liposomes in 4T1 breast cancer-bearing mice.

#### 3.4.3. Melanoma

The drug delivery capacity of albumin-encapsulated liposomes against melanoma was also examined. Zhang et al. compared the antitumor efficacy of chlorambucil-loaded albumin nanoparticle-encapsulated liposomes with chlorambucil-loaded albumin nanoparticles in B16F10 melanoma-bearing mice [76]. The results showed that the growth of the B16F10 melanoma was suppressed in heterotopic B16F10 melanoma-bearing mice that had been treated with chlorambucil-loaded albumin nanoparticle-encapsulated liposomes compared to that of chlorambucil-loaded albumin nanoparticles. The longest survival time was observed in the chlorambucil-loaded albumin nanoparticle-encapsulated liposome group along with an antitumor effect. Another group reported, in an in vitro assay, that paclitaxel-loaded albumin nanoparticle-encapsulated liposomes inhibited the proliferation and migration of B16F10 melanoma cells more strongly than paclitaxel-loaded albumin nanoparticles [51,78]. These superior antitumor effects of albumin-encapsulated liposomes against melanoma may be associated with a higher uptake via energy- and clathrin-dependent endocytosis pathways compared to albumin nanoparticles [76].

#### 3.4.4. Colitis

Okamoto et al. reported on the preparation of tacrolimus-albumin complex-encapsulated liposomes and examined their therapeutic efficacy against colitis induced by dextran sulfate sodium [85]. As a result, the colitis model mice that received the tacrolimus-albumin complex-encapsulated liposomes showed a slightly decreased loss of weight and colon inflammation compared to saline-treated colitis model mice. In addition, albumin-encapsulated liposomes accumulated specifically in the inflammatory site of the colon. These results indicate that albumin-encapsulated liposomes are potent drug carriers for targeting the tumor region and inflammation areas.

## 4. Conclusions

In the past half-century, there have been many successful examples of the use of albumin as a potent biomaterial for biomedicine, including as a carrier to deliver a wide range of drugs as well as bioactive gases to proteins [95,96,97,98,99]. Since these albumin-associated pharmaceuticals and medical preparations have several inherent problems, such as stability, the amount of drug that can be encapsulated, and targeting capacity, these issues may be compensated for by fusing with a liposome. A substantial body of scientific experimental evidence is available to show that albumin-coated liposomes and albumin-encapsulated liposomes are promising carriers for a variety of substances including genes, proteins/peptides, and hydrophobic substances. Such research shows the practical advantages of albumin-associated liposomes over general PEGylated liposomes and albumin-based DDS preparations for the delivery of hydrophobic and hydrophilic substances, the removal of immunogenetics of PEG, improved biocompatibility, and prolonged blood retention. Albumin can compensate for some of the drawbacks associated with the use of liposomes and vice versa. Although preclinical evidence for medical applications of albumin-coated liposomes and albumin-encapsulated liposomes is limited, these albumin-associated liposomes can be used as versatile carriers for treating a variety of solid cancers and inflammatory disorders. Further studies on the albumin in liposome-based DDS are warranted to expand its application as a drug delivery carrier.

## Figures and Tables

**Figure 1 pharmaceuticals-14-00296-f001:**
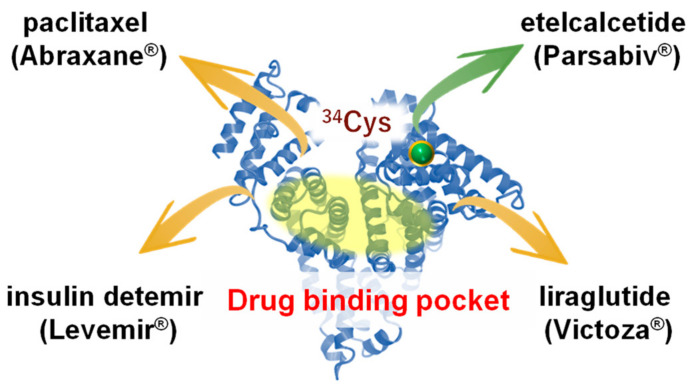
Structural features of albumin and examples of currently marketed albumin-associated pharmaceutical preparations taking advantage of albumin’s innate characteristics.

**Figure 2 pharmaceuticals-14-00296-f002:**
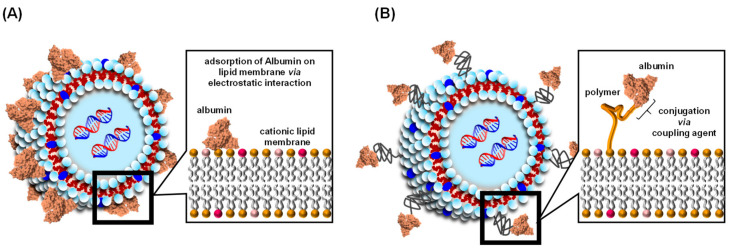
Schematic structure of the liposome coated albumin via electrostatic interaction (**A**) and conjugation using coupling agent (**B**).

**Figure 3 pharmaceuticals-14-00296-f003:**
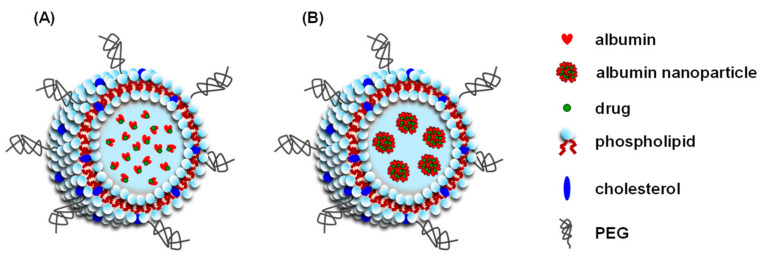
Schematic structure of the albumin-encapsulated liposome-encapsulated albumin-drug complex (**A**) and drug-loaded albumin nanoparticle (**B**).

**Figure 4 pharmaceuticals-14-00296-f004:**
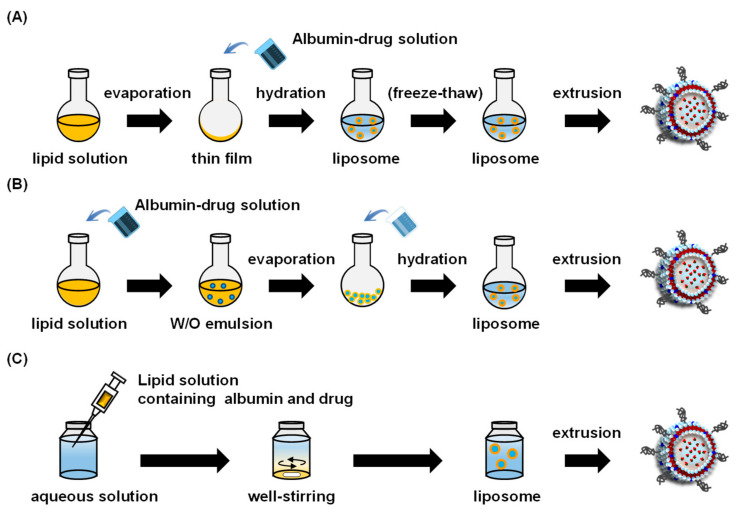
Scheme for the preparation of albumin-encapsulated liposome by thin-film hydration method (**A**), reverse-phase evaporation method (**B**), and ethanol injection method (**C**).

**Figure 5 pharmaceuticals-14-00296-f005:**
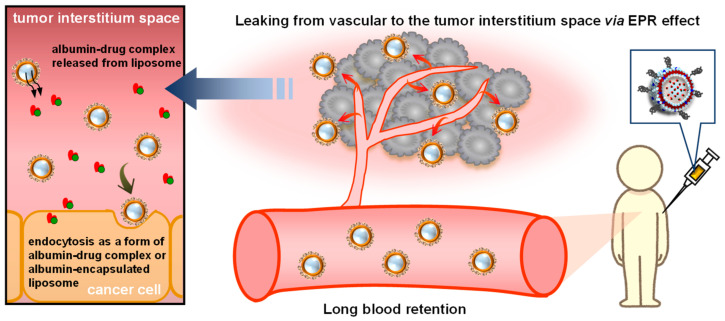
Scheme for the targeting mechanism of albumin-encapsulated liposomes to solid cancers.

**Table 1 pharmaceuticals-14-00296-t001:** List of therapeutic benefits of albumin-encapsulated liposomes for biomedical application.

Target (Cell Line)	Main Therapeutic Results	Ref.
Breast cancer (MCF-7) Melanoma (B16F10)	Paclitaxel-loaded albumin nanoparticle-encapsulated liposome decreased cell viability in 2D-cultured MCF-7 cells and B16F10 cells. No in vivo data was available.	[78]
Melanoma (B16F10)	Chlorambucil-loaded albumin nanoparticle-encapsulated liposomes suppress tumor growth in B16F10 bearing mice with a longer survival duration than chlorambucil-loaded liposomes and chlorambucil-loaded albumin nanoparticles.	[76]
Pancreatic ductal adenocarcinoma (Pan02 and NIH3T3)	Embedding a photothermal agent (IR-780) into the liposomal membrane increased the depth of penetration into the tumor after strong hyperthermal therapy induced by near-infrared laser irradiation. Paclitaxel-loaded albumin nanoparticle-encapsulated liposomes with photothermal therapy dramatically decreased tumor growth and prolonged the survival duration in both heterotopic and orthotopic pancreatic cancer-bearing mice compared to paclitaxel-loaded albumin nanoparticles.	[52]
Breast cancer (MCF-7) Melanoma (B16F10)	Paclitaxel-loaded albumin nanoparticle-encapsulated liposomes containing curcumin decreased cell viability in 2D-cultured MCF-7 cells and B16F10 cells and suppressed B16F10 cell migration in vitro assay. No in vivo data are currently available.	[51]
Breast cancer (4T1)	^131^I-labelled albumin-encapsulated liposome was used for internal isotope therapy. Pretreatment of ^131^I-labeled albumin-encapsulated liposome potentiated the antitumor effect of the second wave of therapies based on biomacromolecules (antibody and liposome) 4T1 breast cancer-bearing mice.	[83]
Breast cancer (MCF-7, MDA-MB-231)	Paclitaxel-albumin complex-encapsulated liposomes decreased tumor growth in a 2D-cultured breast cancer cell and 3D-cultured breast cancer spheroid. No in vivo data are currently available.	[88]
Pancreatic cancer (AsPC-1)	Paclitaxel-albumin complex-encapsulated liposomes decreased cell viability in 2D-cultured AsPC-1 cell. Paclitaxel-albumin complex-encapsulated liposomes suppressed tumor growth in xenograft AsPC-1 bearing mice compared to nab-paclitaxel (Abraxane^®^).	[84]
Pancreatic ductal adenocarcinoma (BxPC-3 and HPaSteC)	Ellagic acid-loaded albumin nanoparticle and paclitaxel-loaded albumin nanoparticle-co-encapsulated liposome increased deep tumor penetration of liposome and suppressed the tumor growth in xenograft BxPC-3 and HPaSteC bearing mice.	[77]
Colitis	Tacrolimus-albumin complex-encapsulated liposome suppressed colon inflammation in a mouse model of colitis induced by dextran sulphate sodium.	[85]

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
