# Peer review of "When Albumin Meets Liposomes: A Feasible Drug Carrier for Biomedical Applications"

_pharmaceuticals, 2021, doi:10.3390/ph14040296_

Round 1
Reviewer 1 Report
Manuscript ID: pharmaceuticals-1158550
When albumin meets liposome: a feasible drug carrier for the biomedical applications
Reviewer(s)’ General Comments to Author:
The paper should be published in the Journal Pharmaceuticals.
There are weak points:
- Authors should emphasize which is practical novelty for such research, and where are/would be applicable these obtained results, in Conclusion for example.
- The Manuscript could benefit from more careful proofreading to eliminate the errors due to poor word usage. The authors should be careful with the typographical things. Several errors have been found throughout the paper.
Author Response
Comment 1:
Authors should emphasize which is practical novelty for such research, and where are/would be applicable these obtained results, in Conclusion for example.
Reply 1
As reviewer suggested, we added sentences regarding practical novelty and applicable examples of albumin-coated liposomes and albumin-encapsulated liposomes in Conclusion of revised manuscript. (line 532-550)
Comment 2:
The Manuscript could benefit from more careful proofreading to eliminate the errors due to poor word usage. The authors should be careful with the typographical things. Several errors have been found throughout the paper.
Reply 2
The revised manuscript has been proofread by native speaker.
Reviewer 2 Report
A very interesting article that reviews all aspects related to the topic covered, from production methods to the different fields of application.
Author Response
Comment:
A very interesting article that reviews all aspects related to the topic covered, from production methods to the different fields of application.
Reply
Thank you for your review.
Reviewer 3 Report
The paper entitled “When albumin meets liposome:a fesible drug carrier for the biomedical applications” by Taguchi et al. reviews the recent advances in the study of albumin-based liposomal formulations as DDS.
The manuscript is well written and clear but it needs to be verified by a native English speaker. Moreover, I suggest the authors to discuss about the preparation of functionalized liposomes for the active drug targeting. Furthermore, the authors must cite a recent review article which discusses the utilization of albumin for the preparation of micellar drug delivery systems: https://doi.org/10.3390/polym13030477
Author Response
Comment 1:
The manuscript is well written and clear but it needs to be verified by a native English speaker.
Reply 1
The revised manuscript has been proofread by native speaker.
Comment 2:
Moreover, I suggest the authors to discuss about the preparation of functionalized liposomes for the active drug targeting.
Reply 2
There is no report that prepares surface modified albumin-associated liposomes for active targeting. However, we agree with your suggestion that discuss the functionalized liposomes for the active drug targeting. Thus, we discussed the possibility of active targeting of albumin-associated liposome in the revised manuscript. (line 307-313)
Comment 3
Furthermore, the authors must cite a recent review article which discusses the utilization of albumin for the preparation of micellar drug delivery systems: https://doi.org/10.3390/polym13030477
Reply 3
As reviewer suggested, we cited the review article written by Atanase. (Reference #: 23)